# Impact of interface traps on charge noise and low-density transport properties in Ge/SiGe heterostructures
Leonardo Massai [1] ✉, Bence Hetényi [1], Matthias Mergenthaler [1], Felix J. Schupp [1], Lisa Sommer [1], Stephan Paredes [1], Stephen W. Bedell [2], Patrick Harvey-Collard [1], Gian Salis [1], Andreas Fuhrer [1] ✉ & Nico W. Hendrickx [1]

Hole spins in Ge/SiGe heterostructures have emerged as an interesting qubit platform with favourable properties such as fast electrical control and noise-resilient operation at sweet spots. However, commonly observed gate-induced electrostatic disorder, drifts, and hysteresis hinder reproducible tune-up of SiGe-based quantum dot arrays. Here, we study Hall bar and quantum dot devices fabricated on Ge/SiGe heterostructures and present a consistent model for the origin of gate hysteresis and its impact on transport metrics and charge noise. As we push the accumulation voltages more negative, we observe non-monotonous changes in the low-density transport metrics, attributed to the induced gradual filling of a spatially varying density of charge traps at the SiGe-oxide interface. With each gate voltage push, we find local activation of a transient low-frequency charge noise component that completely vanishes again after 30 hours. Our results highlight the resilience of the SiGe material platform to interface-trap-induced disorder and noise and pave the way for reproducible tuning of larger multi-dot systems.

Hole spins in germanium quantum dots (QDs)[1–4] are promising qubits for semiconductor-based quantum computing[5]. The intrinsic spin–orbit coupling (SOC) enables fast and local qubit operations[2,4,6,7], with single-qubit gate fidelities well above the fault-tolerant threshold[8]. In particular, strained germanium quantum wells (QWs) have enabled the operation of increasingly larger two-dimensional quantum dot arrays, with demonstrations of four-qubit logic[9], eight-QD analogue quantum simulations[10], and multiplexed addressing of arrays with 16 quantum dots[11]. However, the SOC also induces an interaction between the qubit state and uncontrolled charge fluctuators present in the semiconductor and gate stack[12,13]. Recent work demonstrated that in most regimes of operation, qubit coherence is limited by charge noise[14]. For certain magnetic field orientations, the anisotropic characteristics of heavy-hole-like states[15–18] can enable operational regimes where the sensitivity to noise is suppressed[19–23], but, regardless of the approach chosen to decouple the qubit from noise, reducing charge noise at its source will eventually lead to an enhancement of the overall qubit performance.

Additionally, the reproducible tuning of QD arrays in SiGe heterostructures can be challenging due to local electrostatic disorder[24] and slow drifts[25,26] that depend on the history of the applied gate voltages[27–29]. While this gate voltage-induced hysteresis has been proactively utilized for QD

tuning strategies[30,31], the origins of these phenomena and their impact on the heterostructure properties are still unclear, though crucial for the quality and reproducibility of spin qubits.

In this work, using Hall bar (HB) and QD devices fabricated in the same material stack, we systematically study how transport properties and the charge noise environment of the Ge QW change as we cyclically apply increasingly negative voltages on the accumulation gates. We observe that the device turn-on voltages gradually shift, showing a hysteretic behaviour that depends on the most negative applied gate voltage $V_{min}$. We can attribute this to the induced trapping of holes into a spatially varying density of in-gap traps, located at the SiGe-oxide interface. As the mostly neutrally charged traps get populated by holes tunnelling from the QW to the SiGe-oxide interface, the correspondingly increasing interface charge density and its spatial fluctuations strongly affect the local potential in the QW and, hence the hole gas properties. We measure and compare different two-dimensional transport metrics and find that low-density mobility and percolation density are anti-correlated, while the peak mobility remains unaffected as $|V_{min}|$ is increased. This demonstrates that peak mobility is an inappropriate benchmark for devices operated at low densities, such as spin qubits. Furthermore, by comparing the low-density mobility as a function of $V_{min}$ for Hall bars with different gate oxide thicknesses, we can

[1]IBM Research Europe - Zurich, Säumerstrasse 4, 8803 Rüschlikon, Switzerland. [2]IBM Quantum, T.J. Watson Research Center, 1101 Kitchawan Road, Yorktown Heights, NY, 10598, USA. ✉e-mail: lem@zurich.ibm.com; afu@zurich.ibm.com

quantitatively state that the involved trap ensemble is located at the SiGe-oxide interface. Additionally, by measuring the QD charge noise environment with Coulomb peak tracking as we increase $|V_{min}|$ on the accumulation gate, we observe a ten-fold increase in the low-frequency noise spectrum, caused by an additional Brownian ($1/f^2$) noise component. We attribute this to the local population of interface traps and the activation of relaxation dynamics due to holes that either tunnel back to the QW or relax into neighbouring trap states along the interface. However, while the hysteretic build-up of the electrostatic disorder is persistent unless thermal cycling or illuminating the sample[32], the induced Brownian noise completely vanishes over a timescale of days. After this relaxation period, the original background $1/f$ noise spectrum is recovered, which is likely caused by other sources located e.g. within the gate oxide. Our results explain the origin of the voltage susceptibility and electrostatic variability of SiGe heterostructures with the presence of a spatially inhomogeneous SiGe–oxide interface trap density. They also show that the induced population of traps, while ultimately leading to higher disorder, do not impact the long-term stability and quality of the quantum dots, enabling a hysteresis-aware tuning strategy for large-scale SiGe quantum devices[30,31].

## Results and discussion
### Ge/SiGe heterostructure and device fabrication
We fabricate Hall bar (Fig. 1a) and quantum dot (Fig. 1b) devices on a Ge/SiGe heterostructure. The heterostructure is composed of a strained germanium quantum well (sGe QW) embedded into two silicon–germanium buffer layers and grown using reduced-pressure chemical vapour deposition[33]. The sGe QW is buried 47 nm below the wafer surface, which is capped by a ~1.5-nm-thick oxidized Si layer. Figure 1d shows a transmission electron microscope (TEM) cross-section of the QW region. A schematic illustration of the full gate stack is presented in Fig. 1c. We create ohmic contacts to the QW by annealing Pt into the top SiGe barrier. A first layer of electrostatic gates (GL1, green in Fig. 1c) is defined on top of 7 nm of SiO$_2$ gate dielectric grown by plasma-enhanced atomic layer deposition (PE-ALD). The second layer of electrostatic gates (GL2, blue in Fig. 1c) is separated from GL1 by another 7 nm of SiO$_2$, resulting in a total spacing of

14 nm from the substrate surface. Two types of Hall bar devices are produced using the same fabrication process as the QD devices (see the "Methods" section), with the Hall bar top gate either defined in GL1 (HB$_1$) or GL2 (HB$_2$). The band alignment between the sGe and the SiGe layers defines an accumulation-mode quantum well for holes[34]. When an electric field is applied to the gate electrodes of the device, charges are loaded from the PtSiGe ohmic regions, and a two-dimensional hole gas (2DHG) is accumulated, as shown in the simulation in Fig. 1e (see the "Methods" section).

### Hall bar transport properties
We study the magnetoresistance of Hall bar devices (Fig. 1a) at cryogenic temperatures as a function of the applied top gate voltage. After cooling the device down to ~15 mK, we cyclically repeat the measurement protocol detailed in Fig. 2a (and the "Methods" section). Each measurement cycle starts by first applying an increasingly more negative voltage $V_g = V_{min}$ to the gate, and then stepping $V_g$ from 0 V to $V_{min}$. For every $V_g$ in each cycle, we sweep $B_z$ to measure the Hall carrier density $p$ and Hall transport mobility $\mu$. Furthermore, we extract the percolation density as an alternative benchmark of the hole channel quality.

Focusing on HB$_2$ with an oxide thickness of ~15.5 nm, we first study the impact of hysteresis on the turn-on voltage $V_{t.o.}$. Figure 2b shows all turn-on curves of the channel, for $V_{min}$ decreasing from −0.15 V (red) to −3 V (blue). We define $V_{t.o.}$ as the gate voltage $V_g$ at which the measured longitudinal current $I_{xx}$ reaches 90% of the maximum current ($V_{t.o.} := V_g|_{I_{xx}=0.9I_{xx,max}}$) and plot it as a function of $V_{min}$ in Fig. 2c. The induced shift of $V_{t.o.}$ for negative enough $V_g$ signals the presence of a hysteretic device operation. In particular, we observe five distinct regimes (see the section "Different hysteresis regimes"), delimited by vertical dashed lines:

0. Depleted regime ($−0.13\,V < V_{min}$): channel has not turned on yet;
1. Non-hysteretic regime ($−0.34\,V < V_{min} < −0.13\,V$): channel turn-on voltage $V_{t.o.}$ is independent of $V_{min}$;
2. Screening regime, onset of hysteresis ($−0.5\,V < V_{min} < −0.34\,V$): $V_{t.o.}$ begins to shift with $V_{min}$;

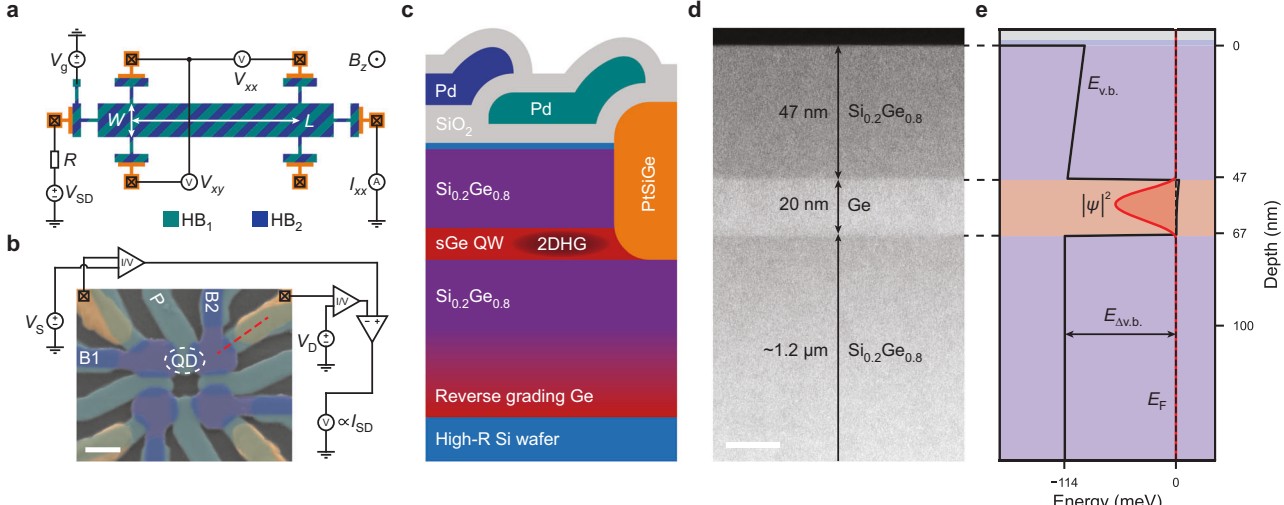

**Fig. 1 | Device layouts and Ge/SiGe heterostructure. a** Schematic illustration of the measurement setup and Hall bars used for magnetoresistance measurements. The Hall bar gate is defined either in GL1 (green) or GL2 (blue) for HB$_1$ and HB$_2$, respectively. Nominally, the channel width is $W = 20$ μm and the length is $L = 100$ μm. We apply a source-drain bias $V_{SD}$ and limit the measured longitudinal current $I_{xx}$ with a serial impedance $R = 10$ MΩ. We measure the longitudinal and Hall voltages, $V_{xx}$ and $V_{xy}$, as a function of the gate voltage $V_g$ and the out-of-plane magnetic field $B_z$. **b** False-coloured SEM-image (following the colour scheme of **c**) of a QD device similar to the one used for the QD measurements. The scale bar is 100 nm. The dashed red line corresponds to the cross-section depicted in (**c**).

We apply source and drain biases ($V_S$ and $V_D$) and measure the differential current $I_{SD}$. **c** Cross-section of the Ge/SiGe heterostructure and gate stack of a QD device. The oxidized Si cap is coloured light blue to distinguish it from the grey PE-ALD SiO$_2$ oxide. **d** Transmission electron microscopy (TEM) image of the sGe QW region. The scale bar is 20 nm. **e** Poisson–Schrödinger simulation of the valence band energy $E_{v.b.}$ (solid black line) in the heterostructure when a negative gate voltage is applied. A 2DHG is accumulated in the sGe QW, (2D density in solid red line). The expected valence band offset between the sGe QW and the SiGe buffer is $E_{\Delta v.b.} \sim 114$ meV[34].

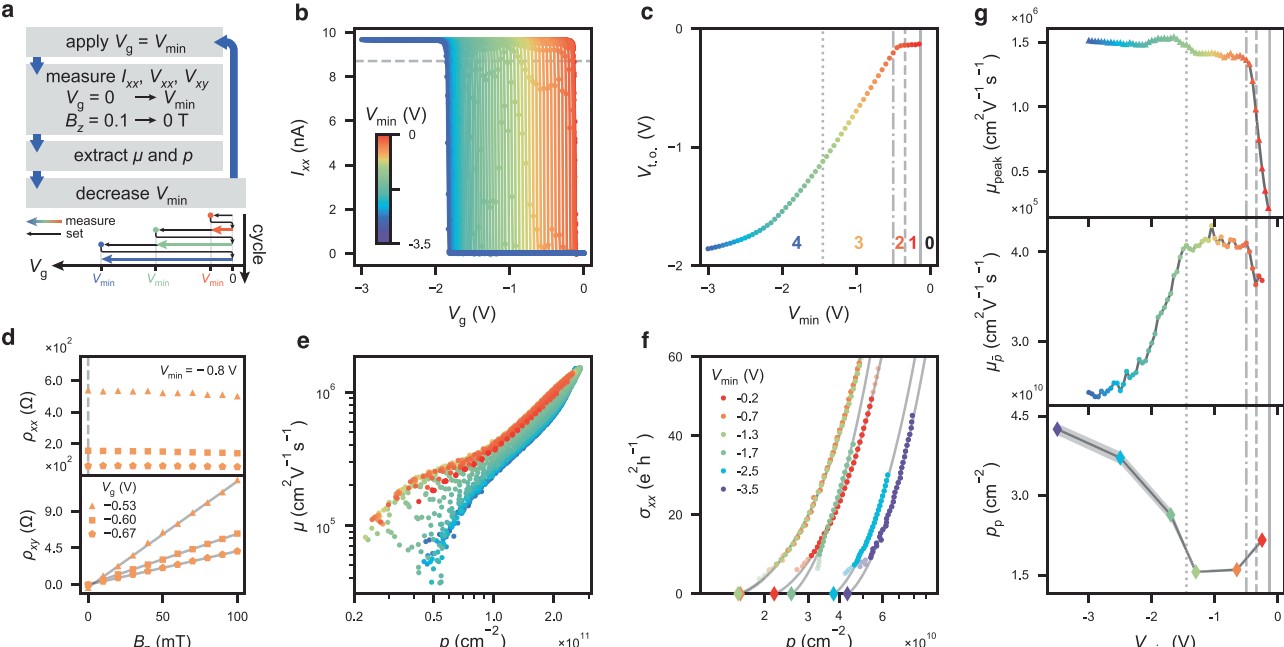

**Fig. 2 | Hall bar measurement data and analysis for HB$_2$. a** Schematic diagram illustrating the measurement protocol. **b** Channel turn-on curves for $V_{min}$ decreasing from $-0.15$ V (red) to $-3$ V (blue). The grey dashed line marks 90% of $I_{xx,max}$, used to extract the turn-on voltage. **c** Extracted turn-on voltage $V_{t.o.}$ as a function of $V_{min}$. The dashed vertical lines separate the different regimes 0–4. **d** Longitudinal resistivity $\rho_{xx}$ (top) and Hall resistivity $\rho_{xy}$ (bottom) as a function of the out-of-plane magnetic field $B_z$, with $V_{min} = -0.8$ V. Different markers represent different $V_g$. The carrier density is extracted from the linear fit to $p(B_z)$ (solid grey lines). **e** Hall mobility $\mu$ as a function of carrier density $p$ extracted for every

$V_{min}$. **f** Longitudinal conductance $\sigma_{xx}$ as a function of $p$, for six different $V_{min}$. The percolation density $p_P$ (diamonds) is extracted by fitting the solid data markers to the percolation theory (see the "Methods" section). **g** Different transport metrics as a function of $V_{min}$: peak mobility $\mu_{peak}$ (top), mobility $\mu_{\bar{p}}$ at low density $\bar{p} = 10^{11}$ cm$^{-2}$ (middle) and percolation density $p_P$ (bottom). The dashed lines separate regimes 0–4. Error bars (shaded area) for $p_P$ are extracted by assessing the stability of the fit when extending the data range to include the transparent markers in **f** (details in the "Methods" section).

3. Linear hysteretic regime ($-1.45$ V $< V_{min} < -0.5$ V): $V_{t.o.}$ shifts proportionally to $V_{min}$;
4. Saturated traps regime ($V_{min} < -1.45$ V): $V_{t.o.}$ asymptotically saturates to a finite value.

We expect the initially measured $V_{t.o.}$ to be determined by multiple factors, such as the already negatively/positively filled traps at the SiGe-oxide interface, fixed charges/dipoles in the gate oxide and the details (strain, composition, layer thickness) of the heterostructure.

Next, we explore the transport properties of the channel in these different regimes. We measure the longitudinal and Hall resistivity, $\rho_{xx}$ and $\rho_{xy}$, respectively, as a function of $B_z$ and $V_g$. Figure 2d shows an example of these data for three different $V_g$ for $V_{min} = -0.8$ V. We extract the mobility-density curve for each $V_{min}$ cycle (see the "Methods" section) as plotted in Fig. 2e. Additionally, we measure the percolation density $p_P$ for six distinct values of $V_{min}$. We extract $p_P$ by fitting the longitudinal conductance $\sigma_{xx}$ at low density to percolation theory[35,36], as plotted in Fig. 2f (fitting procedure in the "Methods" section). We observe a clear change in the mobility–density curve and percolation density as $V_{min}$ is pushed towards more negative values, indicative of a change in the disorder potential impacting the channel. To this end, we extract and compare three different transport metrics (Fig. 2g): peak mobility $\mu_{peak}$ (top, triangles), low-$p$ mobility $\mu_{\bar{p}}$ (centre, dots) at $\bar{p} = 10^{11}$ cm$^{-2}$ and percolation density $p_P$ (bottom, diamonds) as a function of $V_{min}$. The five regimes that we identified in the gate hysteresis behaviour are also reflected in the transport properties (vertical lines) and we will discuss their origin in the subsection "Different hysteresis regimes". The ability to modify the transport properties of the channel by varying $V_{min}$ allows us to compare the different transport metrics. While peak Hall mobility is often used as a key benchmark for heterostructure quality, percolation density $p_P$ is more relevant for quantum materials where isolated charges are accumulated[24,35]. Indeed, we observe

that peak mobility is not representative of the low-density regime, as the trend of $p_P(V_{min})$ is not mirrored by $\mu_{peak}(V_{min})$. Unfortunately, percolation density is more difficult to accurately measure due to the high channel and contact resistances in the low-$p$ regime and the complicated fitting procedure. However, we find that $p_P$ and $\mu_{\bar{p}}$ are strongly anti-correlated as $V_{min}$ is decreased, suggesting that a change in the former can be inferred from a measurement of the latter. We thus propose the mobility at fixed low density as an easy-to-measure metric for benchmarking quantum materials.

**Different hysteresis regimes**

In this subsection, we discuss the origin of the observed regimes in $V_{t.o.}(V_{min})$ and the corresponding features in $\mu_{\bar{p}}(V_{min})$ and $p_P(V_{min})$. Our observations can be explained by the presence of a triangular quantum well (TQW)[27,37,38] in the SiGe barrier above the QW (see Fig. 3a, right panel) and a spatially varying density of neutral in-gap charge traps at the SiGe–oxide interface. The existence of such interface traps is commonly observed in SiGe–SiO$_2$ interfaces[39–41], with typical interface trap densities of $d_{i.t.} \sim 10^{12}$ cm$^{-2}$. The exact physical origin of the charge trapping cannot be determined from the measured data, but potential mechanisms include lattice-mismatch-induced dislocations in the heavily strained Si cap[42] or Ge-rich clusters at the interface[24]. As $V_{min}$ is pushed more negative after the initial cooldown, these traps fill, resulting in a changing charge environment as detected by the transport measurements. Figure 3 details the different processes occurring for the regimes introduced in the section "Hall bar transport properties".

**Regime 0**. The Fermi level of the contacts lies above the highest-energy QW state, such that no charge is accumulated in the device (Fig. 3a, left panel).

**Regime 1**. The Ge QW ground state rises above the Fermi energy of the contacts and a 2DHG is accumulated in the channel. The electric field

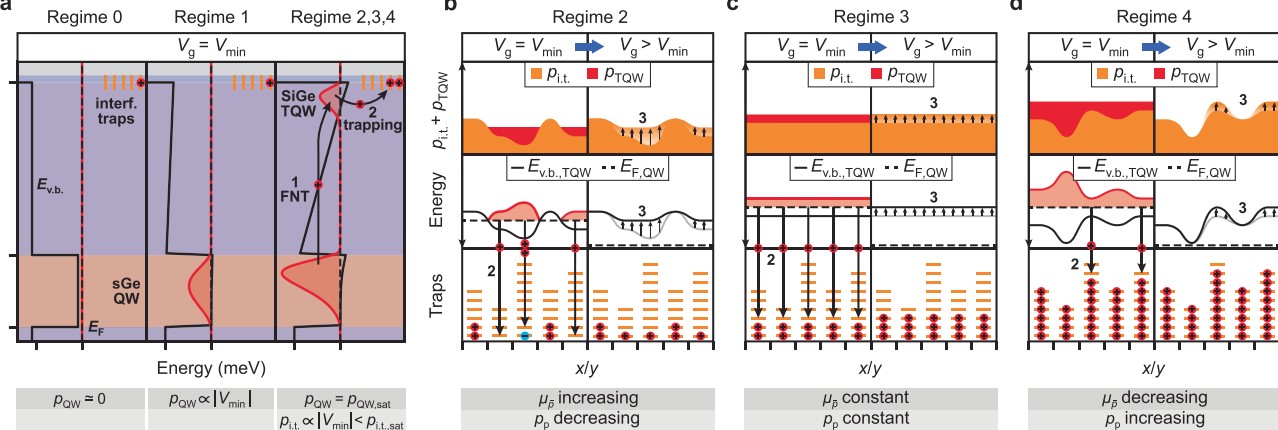

**Fig. 3 | Charge trapping mechanism and effects of a spatially fluctuating density of interface traps. a** Schrödinger–Poisson simulation of the valence band energy (black solid line) and hole density (red line) in regimes 0–4 when $V_g = V_{min}$. Characteristic behaviour of the charge densities in the QW ($p_{QW}$) and trapped at the interface ($p_{i.t.}$) (bottom). **b–d** Illustration of the interface trap distribution and occupation (bottom), the energy diagrams (centre) and charge density at the SiGe–oxide interface $p_{i.t.} + p_{TQW}$ (top) in regime 2 (**b**), regime 3 (**c**) and regime 4 (**d**). Each panel illustrates the spatial charge configuration both when the gate voltage is pushed to $V_g = V_{min}$ (left panels) and during the subsequent low-density measurements (right panels). The black solid line indicates the spatially fluctuating valence band edge in the TQW, $E_{v.b.,TQW}$. The dashed line corresponds to the Fermi energy in the QW, $E_{F,QW}$. Charges tunnel from the QW into the TQW (arrow 1 in **a**, right panel) and subsequently get trapped in the available interface states (arrow 2). This leads to a change in the trapped density $p_{i.t.}$ and the valence band energy $E_{v.b.}$ (arrow 3) acting on the quantum well states. Characteristic behaviour of the low-density metrics as $V_{min}$ is decreased (bottom).

across the SiGe barrier is small enough for the TQW to remain energetically inaccessible and no charge accumulation at the interface occurs (Fig. 3a, central panel) because direct tunnelling between the QW and charge traps can be excluded[37] (see the "Methods" section). As a result, the charge density in the QW increases linearly with the applied gate voltage ($p_{QW} \propto |V_{min}|$) and no hysteresis is observed. While the low-density mobility $\mu_{\bar{p}}$ is independent of $V_{min}$ and initially limited either by fixed charges in the oxide or a spacial variation of the interface charge density after cool down, $\mu_{peak}$ increases with $|V_{min}|$ as a result of improved screening against remote impurity scattering as $p_{QW}$ increases[38].

**Regime 2.** As the electric field strength across the SiGe barrier increases, the TQW starts to be populated by Fowler–Nordheim tunnelling (FNT) from the QW (arrow 1 in Fig. 3a, right panel). From the TQW, charges will get trapped into in-gap interface states (arrow 2 in Fig. 3a, right panel). This accumulation of surface charge lowers the effective electric field across the SiGe barrier and stops the FNT in a self-regulated process[38]. As a result, decreasing $V_{min}$ will lead to an increase of the trapped charge density at the interface, $p_{i.t.}$, while the charge density in the QW stays saturated, $p_{QW} = p_{QW,sat}$ (see Fig. 3b and Supplementary Fig. 1c). Any spatial fluctuations of the valence band edge across the Hall bar, induced e.g. by oxide or interface charges, will lead to a spread of the onset voltages for FNT. This implies that regions with a deeper TQW will get charged more and become less deep, effectively smoothing out the potential fluctuations impacting the QW. The improvement of the low-density mobility and the percolation density with $V_{min}$ can therefore be attributed to a smoothing of the spatially varying disorder potential[43] (Fig. 3b). Regime 2 constitutes the gradual transition between regime 1 (density increasing solely in the QW) and regime 3 (linear increase of the trapped charge at the interface).

**Regime 3.** After the initial disorder potential fluctuations are smoothed, tunnelling to the surface will occur uniformly across the Hall bar (Fig. 3c). The maximum density in the QW is constant throughout this regime and all additional charge gets trapped in the SiGe-oxide interface traps, such that $p_{i.t.} \propto |V_{min}|$. Due to the asymmetric tunnelling rates to the QW and the lack of a mobile channel to the ohmic, these charges remain trapped when the gate voltage is returned to 0 V. As a result, the turn-on voltage

shifts linearly as $V_{min}$ is decreased and $p_{i.t.}$ increases linearly (see the "Methods" section and Supplementary Figs. 1, 2). Transport metrics remain constant throughout this regime and are likely limited by disorder originating in the gate oxide, the QW, or the virtual substrate.

**Regime 4.** As the charge density at the interface increases, all available interface traps are filled, resulting in the accumulation of a finite density $p_{TQW}$ in the triangular quantum well. By comparing the $V_{t.o.}(V_{min})$ data to a one-dimensional Schrödinger–Poisson model, we estimate the density of the interface traps to be $d_{i.t.} \sim 10^{12}\,cm^{-2}$ (see the "Methods" section and Supplementary Figs. 1, 2), in agreement with values measured in similar heterostructures[39–41]. Carriers that tunnel into the TQW can no longer be trapped at the interface, and as $|V_g|$ is reduced, they either tunnel back into the QW or directly into the leads if the percolation threshold in the TQW is reached. Therefore, these carriers do not lead to any further hysteresis. Again, assuming a spatially fluctuating interface trap density, the gate hysteresis gradually saturates as $p_{i.t.} = p_{i.t.,sat} \propto d_{i.t.}$ is reached for different $V_{min}$ across the Hall bar. Furthermore, at low density $p_{QW}$, a fluctuating potential landscape will be present, reflecting the spatially varying interface trap density, which is now highly populated and positively charged (Fig. 3d). This disorder potential will lead to the rapid degradation of the low-density transport metrics as observed in Fig. 2g. Conversely, at high $p_{QW}$, charges loaded into the TQW will offset the interface trap fluctuations such that peak mobility is preserved or even increases slightly with more negative $V_{min}$, as shown in Fig. 4a.

We strengthen our hypothesis by comparing the transition between the different charge loading mechanisms for Hall bars with different gate oxide thicknesses: $HB_1$ and $HB_2$. The transitional regimes (2 and 4) are characterized by a change in low-$p$ mobility $\mu_{\bar{p},HB_i}$, due to spatial fluctuations of the interface quality across the Hall bar, as detailed above. Therefore, to compare the transition voltages for both HBs, we plot $\mu_{\bar{p},HB_i}$ in Fig. 4b and observe that related features in $\mu_{\bar{p}}$ do not appear at the same $V_{min}$ due to the different gate stacks and the corresponding difference in gate capacitance. To quantify the ratio between the transition voltages of each HB, we separate each mobility trace into two parts, isolating the two transitions in the form of abrupt changes in mobility. First, transition I at the onset of FNT (regime 2, yellow in Fig. 4b), corresponding to a steep increase in mobility due to screening of the initial disorder potential. Second, transition II (regime 4, orange in Fig. 4b), corresponding to a decrease in mobility when the

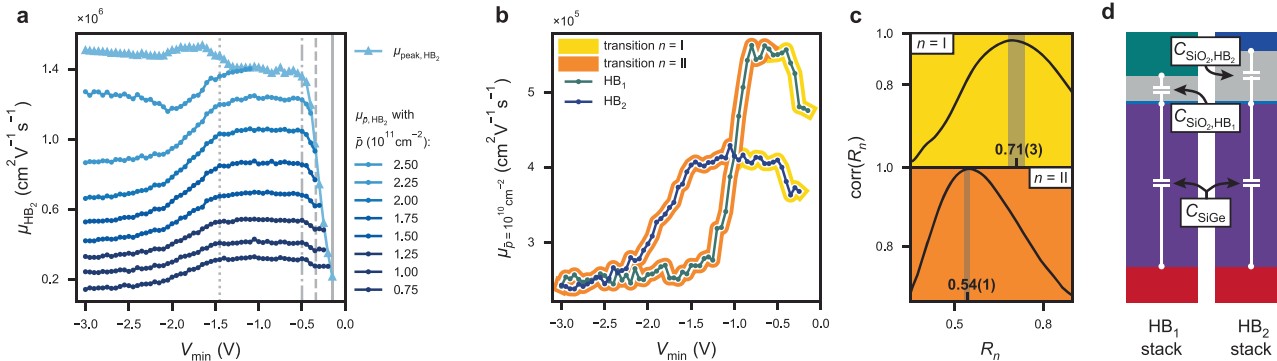

**Fig. 4 | Mobility correlation and gate capacitance study. a** Comparison between the mobility $\mu_{\bar{p}}$ at various fixed densities $\bar{p}$ (circles) as a function of $V_{min}$. The peak mobility $\mu_{peak}$ is also reported (triangles), showing the discrepancy between high- and low-density regimes, particularly for highly negative $V_{min}$. Vertical lines denote the boundaries between the different regimes as defined in the section "Hall bar transport properties". **b** Mobility at low density $\mu_{\bar{p},HB_i}$ measured in $HB_1$ (green dots) and $HB_2$ (blue dots) as a function of $V_{min}$. The yellow and orange shadings indicate the voltage ranges over which the gate-capacitance study is performed for transition I and II, respectively. **c** Correlation between the low-density mobility of each Hall bar as a function of the x-axis scaling factor $R_n$ (see the "Methods" section) for transition $n = $ I (yellow, top) and transition $n = $ II (orange, bottom). The data are plotted for a density $\bar{p} = 10^{11}$ cm$^{-2}$ and the confidence intervals (grey bands) are calculated by repeating the procedure for different densities in the range $\bar{p} = [0.7, 2.0] \times 10^{11}$ cm$^{-2}$ shown in panel (**a**). The maximum correlations are corr($R_I$) = 0.98(2) and corr($R_{II}$) = 0.997(1). **d** Illustration of the gate stack and heterostructure of HB1 (left) and HB2 (right) to scale, including a schematic of the effective planar capacitances between the gate, SiGe–oxide interface, and the QW.

interface traps become fully saturated. Next, we extract the ratio between the transition voltages for each Hall bar, by separately finding the $R_n$ that maximizes corr($R_n$) := corr($\mu_{\bar{p},HB_1}(V_{min}), \mu_{\bar{p},HB_2}(R_n \times V_{min})$) for transition $n = $ I and $n = $ II (see "Methods" section for details). We find $R_I \neq R_{II}$, as shown in Fig. 4c. To explain this difference in the ratio of the transition voltages for both Hall bars, we employ a planar capacitor model, as illustrated in Fig. 4d. When no charge is loaded at the SiGe-oxide interface, the electric field across the SiGe barrier is equal in both Hall bars when the ratio between the applied gate voltages equals $R_{QW} = C_{QW,HB_2}/C_{QW,HB_1}$, with $C_{QW,HB_i}^{-1} = C_{SiGe}^{-1} + C_{SiO_2,HB_i}^{-1}$ being the series capacitance of the SiGe and SiO$_2$ layers. Using nominal layer thicknesses and dielectric constant values from literature[34], we find $R_{QW} = 0.74$. This is in agreement with the extracted voltage ratio $R_I = 0.71(3)$ for transition I, confirming that transition I occurs at a specific electric field in the SiGe barrier. This is consistent with our understanding that the onset of FNT occurs for a specific electric field resulting in a triangular barrier defined by the band offset and depth of the quantum well.

In contrast, near transition II, the electric field across the SiGe is independent on $V_{min}$ as a result of the tunnelling equilibrium between the sGe QW and the SiGe TQW. Decreasing $V_{min}$ only leads to additional charge accumulation at the SiGe–oxide interface and increases the potential drop across the oxide layer. The ratio of gate voltages for which the electric field in the oxide is equal for both Hall bars is determined by the capacitance ratio $R_{SiO_2} \approx C_{SiO_2,HB_2}/C_{SiO_2,HB_1} = 0.55$. This is in agreement with the extracted gate voltage ratio for transition II, $R_{II} = 0.54(1)$, indicating that this transition occurs at a defined electric field in the gate oxide and thus a corresponding fixed charge density at the SiGe–oxide interface, compatible with our understanding of saturating the interface traps.

We also note that by thermal cycling the system from base $T \sim 15$ mK to room temperature and back, the device can be completely reset, which does not happen by sweeping the gate to $V_g = 0$ V (see the "Methods" section). After thermal cycling, the turn-on voltage is reverted to the original value (first red curve in Fig. 2b), indicative of a release of the trapped charges.

**Charge noise**

Next, we perform charge noise measurements on a QD device (Fig. 1b), providing us with a local probe of the charge fluctuators that can limit hole spin qubit coherence[14]. We accumulate a single quantum dot under plunger gate P and observe clean, regular Coulomb peaks (CPs) in the measured source-drain current $I_{SD}$ (Fig. 5a). In addition, gates B1 and B2 can be used to control the tunnel coupling to the source and drain reservoirs,

respectively. To observe the effects of gate hysteresis on charge noise, we employ a similar measurement protocol as for the Hall bars, where we measure the charge noise as we cyclically push the plunger gate voltage to more negative $V_{min}$, as detailed in Fig. 5a. After pushing the plunger gate voltage $V_P$ to $V_{min}$, we tune $V_P$ to locate the first measurable CP at $V_{P,CP}$ (see the "Methods" section and Supplementary Fig. 3a) and observe a hysteretic behaviour with $V_{P,CP}$ shifting linearly with $V_{min}$. Next, we assess the charge noise, using the Coulomb peak tracking (CPT) method, where $V_P$ is repeatedly and synchronously swept across the CP. This method allows us to probe very low-frequency noise, and we track the CP position $V_{P,CP}(t)$ for $t = 1.5$ hours by fitting the individual traces to a Gaussian function, as shown in Fig. 5b (see the "Methods" section). The CP position fluctuates over time, as a result of nearby charge fluctuators capacitively coupled to the QD.

In Fig. 5c, we compare $V_{P,CP}(t)$ for different $V_{min}$ and find that the amplitude of the fluctuations increases for more negative $V_{min}$. To quantify this effect, we take the fast Fourier transform of $V_{P,CP}(t)$ and extract the power spectral density (PSD) $S_V$ for each $V_{min}$. Using the plunger gate lever arm $\alpha_P \approx 0.23$ (see Supplementary Fig. 3b), we convert the PSD onto an energy scale and extract the noise spectral density $S_E^{1/2}$ at $f = 10^{-2}$ Hz (Fig. 5d). As $V_{min}$ is decreased, the low-frequency noise $S_{E,f=10^{-2}Hz}^{1/2}$ increases over an order of magnitude and then saturates similarly to the low-density transport metrics. The observed trend of increasing noise and reduced stability of the Coulomb peaks is likely also linked to the filling of the SiGe–oxide interface traps. To get a better insight into the underlying physical mechanism, we fit every PSD trace $S_V$ over the measured frequency range to a power law $S_0/f^\alpha$ and compare the noise exponents $\alpha$. We find that $\alpha$ increases from ~1.4 to ~1.8 as $V_{min}$ is pushed more negative (Supplementary Fig. 4). A deviation from the expected $1/f$ PSD can be caused by a few fluctuators interacting strongly with the quantum dot[44] or a noisy relaxation process that leads to an Ornstein–Uhlenbeck behaviour[45]. In our case, we observe that the CP position exhibits a noisy drift that increases with $V_{min}$ and adds a low-frequency $1/f^2$ component to the underlying $1/f$ noise spectrum, despite letting the system settle for ~ 10 min after pushing $V_P$ to $V_{min}$. We believe that this charge offset drift[25], with a Brownian PSD, is caused by the slow relaxation of the charges accumulated at the interface, as a result of low tunnel rates to nearby charge traps or back to the QW.

In the penultimate measurement cycle ($V_{min} = -2.05$ V, diamond data point in Fig. 5d), we investigate how this low-frequency noise evolves over time. We extract $S_{E,f=10^{-2}Hz}^{1/2}$ as a function of the waiting time $T$ after setting $V_P$ to $V_{min}$ and repeatedly take 2-hour-long CPT measurements over

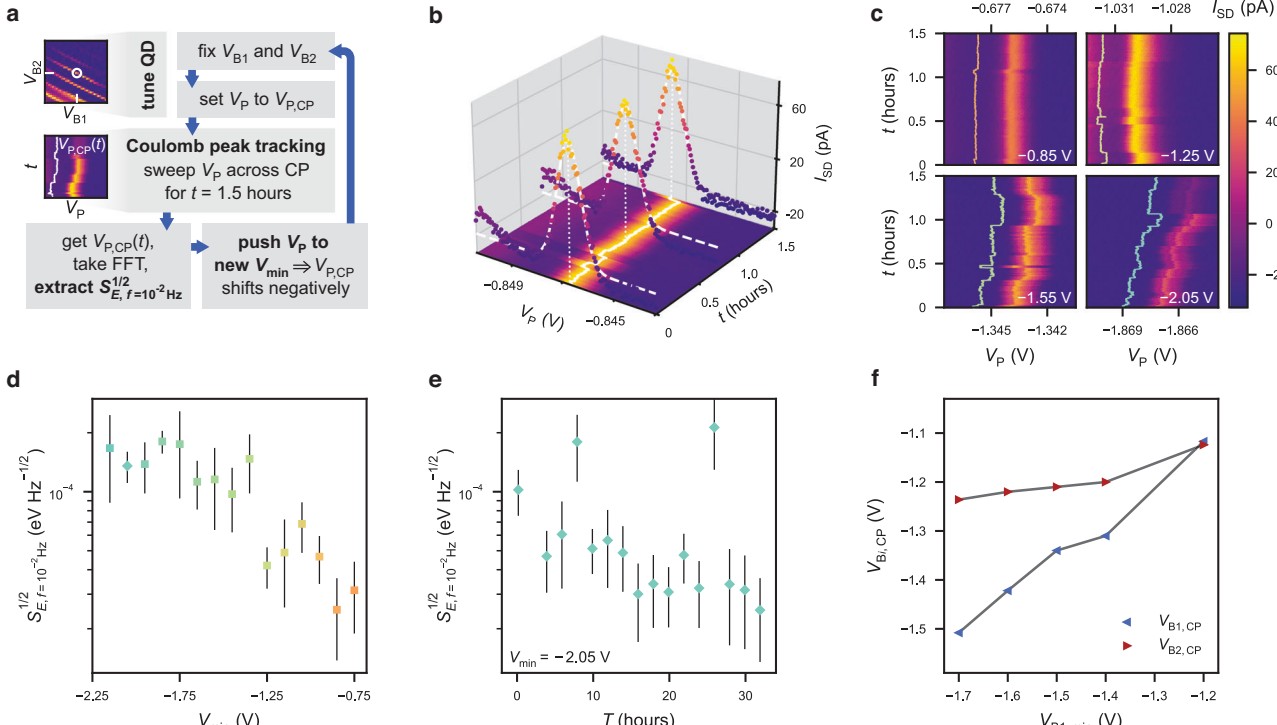

**Fig. 5 | Charge noise measurement and data analysis in QD. a** Schematic illustration of the measurement protocol. We repeatedly perform charge noise measurements using the CPT method, as the plunger gate voltage is pushed more negatively. **b** CPT charge noise measurement for $V_{\min} = -1.05$ V, highlighting three $I_{SD}$ traces (dots) and the respective Gaussian fits (dashed white lines). The solid white line indicates the extracted $V_{P,CP}(t)$. **c** CPT charge noise measurements for different $V_{\min}$ values (white text). The solid line illustrates the extracted $V_{P,CP}(t)$, offset by $-2$ mV for visibility. **d** Charge noise spectral density $S^{1/2}_{E, f=10^{-2} Hz}$ extracted from the measured $S_V$ as a function of $V_{\min}$. The central values and the corresponding error bars plotted at each $V_{\min}$ are, respectively, the means and the standard deviations of the noise in a frequency interval of $\pm 5\%$ around $f = 10^{-2}$ Hz. **e** Charge noise spectral density $S^{1/2}_{E, f=10^{-2} Hz}$ as a function of time $T$ passed since the $V_P$ has been set to $V_{\min} = -2.05$ V (diamond marker in **d**). The plotted central values and the corresponding error bars are obtained as in **d**. **f** Voltage set on B1 ($V_{B1,CP}$, blue) and B2 ($V_{B2,CP}$, red) to stay on the CP resonance with symmetric reservoir tunnel rates ($V_P = -0.4$ V) as the voltage on B1 is pushed to $V_{B1,\min}$ in cycles.

a time span of >30 hours (full data in Supplementary Fig. 5). The results are shown in Fig. 5e, and we observe that the low-frequency noise intensity decreases monotonously, approaching the lowest noise level measured initially at $V_{\min} = -0.75$ V. The remaining background $1/f$ noise component is therefore ascribable to other sources. The two outliers in Fig. 5e are caused by a large jump of the CP position, $V_{P,CP}(t)$ during the CPT measurement. We then confirm that the increase in noise is gate voltage-induced and reproducible by pushing $V_P$ to $V_{\min} = -2.15$ V and acquiring the leftmost data point in Fig. 5d. The charge noise increases to an intensity similar to the previous cycle. Since the characteristic time scale of the noise decay is of the order of a day, the increased noise power is visible only at very low frequencies ($f < 10^{-2}$ Hz) and cannot easily be observed using e.g. the Coulomb peak flank (CPF) method (see the "Methods" section).

Additionally, in a separate cool down, we fix $V_P = V_{P,CP}$ and cyclically push the voltage on barrier gate B1 to increasingly negative voltages $V_{B1,\min}$. After each cycle, we tune $V_{B1}$ and $V_{B2}$ to recover similar and symmetric tunnel rates (Supplementary Fig. 3c). We observe that this predominantly requires a gate voltage correction on gate B1, as shown in Fig. 5f. This shows that the charge trap filling is a local effect, arising close to the pushed gate, and thus confirms that voltage hysteresis and noise are linked to charge traps at the SiGe–oxide interface rather than defects deeper down in the heterostructure stack.

## Conclusions

We studied and modelled the voltage-induced hysteresis commonly observed in SiGe heterostructures. We pinpoint its origin to the population of distribution of neutral charge traps with varying densities along the SiGe–oxide interface. By applying increasingly negative gate voltages, holes gradually tunnel from the QW to the interface, where they get permanently trapped. A full reset is possible by thermally cycling the device or by illumination, but not just by returning the gate voltage to 0 V. The population of these interface traps has multiple effects. First, it defines different operation regimes with corresponding levels of static disorder, as probed by the transport metrics in HBs. Initially, the population of charge traps can help to smooth a built-in disorder potential, leading to an improvement in transport metrics. However, as the trap density is locally saturated, the trapping along the interface becomes nonuniform, reintroducing and increasing disorder. This particularly affects the hole gas in the few-carrier regime, and as a result we find that both the low-density mobility and the percolation density are fully anti-correlated as a function of the most negative applied gate voltage, $V_{\min}$. In contrast, we observe that the peak mobility, which is found at peak density, remains mostly unchanged as the fluctuations in trapped charges are smoothed by the population of the triangular quantum well at the interface. This illustrates that peak carrier mobility is not a good benchmark for quantum materials and devices. Instead, we propose the mobility at fixed low density as an easy-to-measure and relevant benchmark. Second, using a QD as a probe of the local electric field, we confirm that the charge traps are populated in the proximity of the gate whose voltage was pushed. Furthermore, charge noise measurements show a strong transient $1/f^2$ component at low frequencies due to electrically activated noisy relaxation dynamics that result from slow rearrangements of trapped charges as the system reaches equilibrium. This transient noise component fades away over a timescale of a few days, recovering an underlying $1/f$ spectrum, which we ascribe to other sources, e.g. in the oxide. Our results highlight the importance of the SiGe–oxide interface quality for the realization of reproducible and homogeneous Ge quantum devices, and the detailed understanding of the charge trapping mechanism allows the identification of different operation regimes. Furthermore, our results

enable extensible tuning strategies that rely on the observed turn-on voltage hysteresis without compromising the long-term qubit stability and ultimately show the resilience of the Ge/SiGe heterostructure to trap-provoked noise dynamics.

## Methods

### Device fabrication

The Hall bar and quantum dot devices are fabricated on a Ge/SiGe heterostructure as depicted in Fig. 1c of the main text[33]. The ohmic contacts to the QW are realized by evaporating 50 nm of Pt/Ti on the heterostructure and diffusing the metal into the top SiGe barrier at a temperature of 300 °C for about 1 hour during the gate oxide deposition, resulting in the formation of PtSiGe. We note that in the devices used throughout this work, the Pt-silicide-germanide did not reach the QW, leading to a large contact resistance (~MΩ for the QD device). Electrostatic gates are then defined using electron beam lithography and lift-off of Ti/Pd (20 nm), separated by thin (7 nm) layers of $SiO_2$ deposited at 300 °C via plasma-enhanced atomic layer deposition (PE-ALD). The first (second) gate layer GL1 (GL2), coloured green (blue) in Fig. 1c, has a total of ~1.5 + 7 = 8.5 nm (~1.5 + 7 + 7 = 15.5 nm) of $SiO_2$ gate oxide including the oxidized Si cap. We note that no special surface treatment is performed before depositing the first $SiO_2$ layer, aside from standard acetone/IPA cleaning.

### Experimental setup

The sample, mounted on a QDevil QBoard circuit board, is loaded in a Bluefors LD400 dilution refrigerator and cooled down to a base temperature of $T \approx 10$ mK.

For the Hall bar magnetoresistance measurements, we use three lock-in amplifiers (Signal Recovery 7265) with 12 dB/oct filters. With lock-in amplifier #1, we generate an oscillating bias voltage (amplitude $V_{RMS} = 0.1$ V, frequency $f_0 = 3$ Hz) that is applied to the Hall bar source contact through a 10 MΩ resistor, defining an effective current source when the channel is sufficiently open. A Basel Precision Instruments (BasPI) SP983c IV-converter (gain = $10^8$, $f_{cut-off} = 300$ Hz) is connected to the HB drain contact, and the bias current $I_{xx}$ is measured using lock-in amplifier #1. We directly extract the differential longitudinal and Hall voltages $V_{xx}$ and $V_{xy}$ using two BasPI SP1004 differential voltage amplifiers (gain = $10^3$, $f_{cut-off} = 300$ Hz) connected to lock-in amplifiers #2 and #3, respectively (both synchronized to lock-in amplifier #1). The dc gate voltage $V_g$ is applied to the HB gate using a QDevil QDAC through twisted-pair wiring and filtered using a QDevil QFilter at the millikelvin stage of our fridge. The out-of-plane magnetic field $B_z$ is applied by an American Magnetics three-axis magnet with a maximum field of 1/1/6 Tesla in the x/y/z direction and a high-stability option on all coils. For the charge noise measurements, the quantum dot device is dc-biased using a BasPI low noise high-resolution DAC II. We apply a source–drain bias excitation of $V_{SD} = 300$ μV and measure the differential current $I_{SD}$ using a pair of BasPI SP983c IV-converters and a SP1004 differential amplifier (gain = $10^3$, $f_{cut-off} = 300$ Hz) connected to a Keysight 34461A digital multimeter.

### Hall bar measurement protocol

Here, we detail the cyclic measurement protocol used for the HB transport measurements, as illustrated in Fig. 2a. Initially, the device is reset by performing a thermal cycle to room temperature. At the start of every measurement cycle, the HB gate voltage is swept (at a rate of 1 V s$^{-1}$) to $V_g = V_{min}$. Next, the gate voltage is left at $V_g = V_{min}$ for a waiting period of $t_{wait} = 60$ s after which $V_g$ is swept back to 0 V ($t_{wait} = 0.5$ s). Subsequently, the longitudinal current $I_{xx}$, voltage $V_{xx}$ and Hall voltage $V_{xy}$ are measured as $V_g$ is swept from 0 V to $V_{min}$ and $B_z$ is stepped from 100 to 0 mT. The measurement is repeated in cycles, decreasing $V_{min}$ in steps of $\delta V_{min} = 50$ mV, with the measurement range of $V_g$ increasing correspondingly. The percolation density measurements are performed in a separate cooldown, following a similar cyclic approach. For the $p_p$ measurements at low $V_{min} < -2$ V, a longer waiting time was introduced, keeping $V_g = V_{min}$ for ~5 min, to let the channel turn-on curve stabilize.

### Device reset

The induced population of the interface traps can be fully reset by thermal cycling the device to room temperature. We performed this procedure multiple times and observed that the measured properties, e.g. turn-on voltage as a function of $V_{min}$, are reproducible over multiple cooldowns. Other recent works explored alternative reset methods, such as applying strong voltages with opposite signs on the accumulation gate[31] or illuminating the device with optical light[32]. We tested both approaches, achieving similar results but not consistently resetting the device to the original state, leaving this to future investigations.

### Extraction of charge-carrier density and mobility

Figure 2d shows the measured longitudinal and Hall resistivities, $\rho_{xx}$ and $\rho_{xy}$, respectively, for one of the measurement cycles ($V_{min} = -0.8$ V). By fitting $\rho_{xy} = B_z/ep + c$, with $e$ the elementary charge, we can extract the classical density $p$ ($c$ is a small offset value, added to account for possible offsets in $V_{xy}$ when $V_g \approx V_{t.o.}$). Consequently, the classical mobility is calculated as $\mu = 1/ep\rho_{xx}|_{B_z=0}$. Mobility vs. density for each cycle is plotted in Fig. 2e. From this, it is possible to extrapolate the mobility at fixed density, as shown in Fig. 4a.

### Extraction of percolation density

Percolation density measurements are performed on $HB_2$ after resetting the interface traps by thermal cycling the device. As the channel and contact resistance at low density is larger, we bias the device through a 100 MΩ resistor (see Fig. 1a), with $V_{RMS} = 1$ V, $f_0 = 3$ Hz, maintaining $I_{xx,max} \sim 10$ nA. Since direct measurement of the channel percolation density, $p_p$ is challenging due to the increasing (contact) resistance, we extract $p_p$ from the longitudinal conductance $\sigma_{xx}$ (measured down to ~5 $e^2\,h^{-1}$) by fitting it to $\sigma_{xx} \propto (p - p_p)^{1.31}$, as expected from percolation theory[35,36,46]. Figure 2f shows the measured $\sigma_{xx}$ as a function of charge-carrier density $p$. The error bars of the percolation density (shaded area in Fig. 2g) account for both the maximal uncertainty arising from the non-linear least square fit and the spread in $p_p$ observed when including/excluding the transparent data points shown in Fig. 2f from the fitting.

### HB gate capacitance

In ref. 37, direct tunnelling from the QW to the interface traps was found to strongly influence the slope of the measured Hall density as a function of gate voltage $p(V_g)$, leading to a reduced effective gate capacitance. Here, to rule out direct tunnelling between the QW and interface traps as a significant trap-filling mechanism, we perform the same analysis. The nominal gate capacitance, used for the comparative analysis of the mobility in the main text, is calculated as $C_{QW,HB_i}^{-1} = C_{SiGe}^{-1} + C_{SiO_2,HB_i}^{-1}$ resulting in $C_{QW,HB_1,nom} = 3.38$ pF and $C_{QW,HB_2,nom} = 2.52$ pF using the values reported in Table 1. By linearly fitting the unsaturated part of $p(V_g)$, we extract $C_{QW,HB_1,meas} = 3.04(2)$ pF and $C_{QW,HB_2,meas} = 2.40(1)$ pF. We attribute the small differences (~9% and ~4%, respectively), to inaccuracies in the used layer thickness and dielectric constant values, compared to the >80% discrepancy observed in ref. 37 attributed to direct tunnelling.

### Mobility correlation analysis

As discussed in the main text, we extract the ratio $R_n = V_{n,HB_1}/V_{n,HB_2}$ between the voltages at which the transition $n \in [I, II]$ occurs for each HB. The transitions are characterized by a change in the low-density ($p = \bar{p}$) mobility, allowing us to find $R_n$ by calculating the Pearson product-moment correlation coefficient between the $V_{min}$-dependence of the mobility $\mu_{\bar{p}}(V_{min})$ for each HB:

$$
\begin{aligned}
\mathrm{corr}\,(R_n) \quad &:= \quad \mathrm{corr}\left(\mu_{\bar{p},HB_1}\left(V_{min}|_{V_{n,a}}^{V_{n,b}}\right), \mu_{\bar{p},HB_2}\left(R_n \times V_{min}|_{V_{n,a}}^{V_{n,b}}\right)\right) \\[2mm]
&= \quad \frac{\sum_{V_{min}=V_{n,a}}^{V_{n,b}} (\mu_{\bar{p},HB_1}(V_{min}) - \bar{\mu}_{\bar{p},HB_1})(\mu_{\bar{p},HB_2}(R_n \times V_{min}) - \bar{\mu}_{\bar{p},HB_2})}{\sqrt{\sum_{V_{min}=V_{n,a}}^{V_{n,b}} (\mu_{\bar{p},HB_1}(V_{min}) - \bar{\mu}_{\bar{p},HB_1})^2} \sqrt{\sum_{V_{min}=V_{n,a}}^{V_{n,b}} (\mu_{\bar{p},HB_2}(R_n \times V_{min}) - \bar{\mu}_{\bar{p},HB_2})^2}}
\end{aligned}
$$

$$(1)$$

**Table 1 | Gate structure values used for gate capacitance**

| Layer | Material | $\epsilon_r$ | $t$ (nm) |
|---|---|---|---|
| Gate ox. 2 (HB$_2$) | SiO$_2$ | 3.9 | 7.0 |
| Gate ox. 1 | SiO$_2$ | 3.9 | 7.0 |
| Oxidized Si cap | SiO$_2$ | 3.9 | 1.5 |
| Top barrier | Si$_{0.2}$Ge$_{0.8}$ | 15.34 | 47.0 |

Material, dielectric constant and thickness of each gate stack layer are used to calculate the nominal gate capacitance. The dielectric constants are extracted or interpolated (Si$_{0.2}$Ge$_{0.8}$) from ref. 34.

**Table 2 | $V_{min}$ domains for the capacitance analysis**

| HB$_i$ | $V_{I,a}$ (V) | $V_{I,b} = V_{II,a}$ (V) | $V_{II,b}$ (V) |
|---|---|---|---|
| HB$_1$ | 0 | −0.57 | −3 |
| HB$_2$ | 0 | −0.95 | −3 |

We define two voltage domains between $V_{n,a}$ and $V_{n,b}$, corresponding to transition $n \in$ [I, II] of each HB. The two voltage domains share one boundary ($V_{I,b} = V_{II,a}$).

where $[V_{n,a}, V_{n,b}]$ defines the voltage range of transition $n$ as shown in Fig. 4b and reported in Table 2. $\bar{\mu}_{\bar{p},HB_i}$ is the mean low-$p$ mobility in the voltage range of transition $n$ for Hall bar $i \in$ [1, 2].

**Coulomb peak tracking method**

We define the effective Coulomb peak potential $V_{CP}$ (shown in Supplementary Fig. 3a) as

$$V_{CP}(V_{min}) = V_{P,CP}(V_{min}) + \sum_{i=1,2} \alpha_{P,Bi} V_{Bi,CP}(V_{min}) \qquad (2)$$

where $V_{P,CP}$, $V_{B1,CP}$ and $V_{B2,CP}$ are the voltages set respectively on gates P, B1, and B2 to be on Coulomb resonance. $\alpha_{P,Bi}$ is the relative capacitance of B$i$ with respect to P ($\alpha_{P,B1} = 0.18$, $\alpha_{P,B2} = 0.31$). For every cycle of the measurement, after pushing $V_P$ to the new $V_{min}$ and waiting for 1 min, $V_P$ is swept to locate the first measurable Coulomb peak at $V_{P,CP}$. The voltage on the barriers is kept approximately constant throughout the experiment, with only small corrections ($\Delta V_{Bi,CP} \sim 0.1$ V) to ensure a measurable current level. The CP is then used to extract the charge noise value.

We measure charge noise using the Coulomb peak tracking (CPT) method. This method differs from the more commonly used Coulomb peak flank (CPF) method[35,47], where $V_P$ is fixed on the CP flank, and current fluctuations are measured over time. Using CPT, the highest noise frequency $f_{high}$ that can be extracted is limited by the duration of a single $V_P$ sweep. This is ultimately limited by the current integration time (20 ms) and by the number of points per voltage sweep (150), leading to a sweep time of $t_{sweep} = 150 \times 20$ ms + 1 s = 4 s for our measurements (a waiting time of 1 s is added to reset the triggering since subsequent sweeps need to be synchronous). The lowest measurable noise frequency $f_{low}$, however, is set by the total measurement duration $t$ (1.5 hours for our measurements), as long as the CP remains within the measurement window. In contrast, for the CPF method, $f_{high}$ is only defined by the integration time, typically resulting in a much larger $f_{high}$. However, this method requires the CP to remain in an approximately linear part of the flank. When the CP moves by a $\delta V$ large enough such that this requirement is broken, the measurement is effectively terminated. This typically limits $f_{low}$, with measurement times longer than several minutes being difficult to achieve. Using CPT, we are thus less sensitive to high-frequency noise but are able to measure down to very low frequencies.

**Band structure simulations**

We use a one-dimensional self-consistent Schrödinger–Poisson solver[48] to obtain the band structure and hole densities in our Ge/SiGe heterostructures. Parameters of the SiGe band structure are extracted from ref. 34, while for some parameters like the dielectric constant, the SiGe value was obtained from linear interpolation. Here, we focus on three regimes: the regular conductance in the Ge channel without hysteresis (regime 1 in the main text), the onset of the accumulation at the interface due to Fowler–Nordheim tunnelling[49] (regime 2), and the linear hysteretic regime, where the turn-on voltage is shifted by charge trap filling at the SiGe–oxide interface (regime 3). Our simulations have been performed at $T_{sim} = 10$ K in order to avoid numerical instabilities, but we note that the thermal occupation of states is negligible in the relevant range of densities. Furthermore, the effective 1D simulation returns the hole density in thermal equilibrium. Therefore discrepancies are to be expected due to non-equilibrium processes as well as the deviation between charge density and Hall-density[50].

Let us first consider the Ge/SiGe heterostructure at small negative gate voltages where the Fermi energy lies inside the band gap close to the edge of the valence band. In Supplementary Fig. 2a we see that by applying a gate voltage of $V_g = -0.15$ V, the channel starts to accumulate holes, introducing a net electric field in the SiGe layer. Note that the negligible charge density $p_{TQW}$ in the SiGe buffer layer is a result of thermal occupation, which should be suppressed even further in the experiment ($T_{sim} = 10$ K, whereas $T_{exp} \sim 15$ mK).

Decreasing the gate voltage further to $V_g = -0.35$ V, to the onset of hysteretic regime (regime 2 in Fig. 2c), the tip of the band edge of the SiGe layer reaches the band edge of the channel and carriers start to accumulate in the SiGe by means of Fowler–Nordheim tunnelling from the Ge channel[49] (Supplementary Fig. 2b). In this regime, the charge density in the QW has reached a saturation density $p_{QW,sat} = 1.7 \times 10^{11}$ cm$^{-2}$, determined by the condition that the voltage drop between QW and the oxide equals to the band-gap mismatch. The obtained saturation density $p_{QW,sat}$ is significantly lower than the maximal density measured in this regime (see Supplementary Fig. 1c). The disparity can come from the parameters used in the simulation, such as the width of the SiGe layer, the dielectric constant or the band-gap mismatch between the SiGe and the QW. Alternatively, ref. 43 argues that non-equilibrium processes can explain such deviations due to the Fermi-level pinning near the oxide interface, which could allow for slow tunnelling into the interface traps even before the conditions for Fowler–Nordheim tunnelling are met.

In order to reproduce the large shift of the turn-on voltage, we assume that the total charge density accumulated in the triangular well (e.g. $p_{TQW} = 1.5 \times 10^{12}$ cm$^{-2}$ for $V_{min} = -1.45$ V as in Supplementary Fig. 2c) remains trapped at the interface when the gate voltage is swept back to zero. As the exact location of the trapped charges is unknown, we assume in the simulation that they are uniformly distributed in the oxidized Si cap layer between the SiGe buffer and the gate oxide. We note that the charge densities we find are comparable to literature values of the interface trap density in SiGe–SiO$_2$ interfaces, e.g. $d_{i.t.} \sim 10^{12}$ cm$^{-2}$ as measured in ref. 39. The charge traps are filled when the gate voltage is initially set to $V_{min}$ at the beginning of each cycle. When the gate voltage is subsequently swept from 0 V to $V_{min}$ during the transport measurement, the charge accumulation in the TQW is highly reduced due to the repulsion of the trapped charges ($p_{TQW} = 1.3 \times 10^{11}$ cm$^{-2}$), as can be seen in Supplementary Fig. 2d. The fact that the carrier density at the interface does not drop to zero is a consequence of fixing the corresponding charge density to the ~1 nm-thick layer above the SiGe layer instead of its equilibrium distribution shown in Supplementary Fig. 2c. As a result, the reminiscent carrier density in the TQW in the simulation is not necessarily representative for the densities observed in the experiment, as it strongly depends on the location of the charge traps that is not taken into account in the simulation.

Finally, we extract the turn-on voltage shift as a result of the above-described complete charge trapping process and plot this in Supplementary Fig. 1a and b for HB$_1$ and HB$_2$, respectively. The simulated $V_{t.o}$ is in good agreement with the observed turn-on voltage shift of both Hall bars, supporting our understanding that all surface charge initially remains trapped.

## Data availability

All data underlying this study are available at Zenodo at https://doi.org/10.5281/zenodo.8410963 [51].

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

## Acknowledgements

We acknowledge the staff of the Binnig and Rohrer Nanotechnology Center for their contributions to the sample fabrication and thank all members of the IBM Research Europe—Zurich spin qubit team for useful discussions. We thank Gregory Snider for providing access to the updated 1D Schrödinger–Poisson solver. We acknowledge funding from the NCCR SPIN under SNSF grant no. 51NF40 180604, and A.F. acknowledges support from the SNSF through grant no. 200021 188752.

## Author contributions

L.M. performed the experiments and data analysis with the help of N.W.H.; B.H. performed the simulations; L.M., B.H., A.F., and N.W.H. developed the theoretical model. N.W.H. fabricated the sample with contributions from M.M., F.J.S. and L.S.; S.P., P.H.-C., and G.S. contributed to the development of the experimental setup. S.W.B. provided the heterostructures. L.M. wrote the manuscript with contributions from B.H., A.F., and N.W.H. and input from all authors; A.F. and N.W.H. supervised the project.

## Competing interests

The authors declare no competing interests.

## Additional information

**Peer review information** *Communications Materials* This manuscript has been previously reviewed at another Nature Portfolio journal. The manuscript was considered suitable for publication without further review at Communications Materials.

