## [Peer Review File · Communications Materials]

This manuscript has been previously reviewed at another Nature Portfolio journal. The manuscript was considered suitable for publication without further review at *Communications Materials*.

23rd May 24

Dear Mr Massai,

Thank you for transferring your manuscript, "Impact of interface traps on charge noise, mobility and percolation density in Ge/SiGe heterostructures", to Communications Materials. Based on the referees' reports and your previous replies, I am delighted to say that we are happy, in principle, to publish a suitably revised version in Communications Materials under the open access CC BY license (Creative Commons Attribution v4.0 International License). Please accept our apologies for the delay getting back to you, caused by one of our editors currently being away.

We therefore invite you edit your manuscript to comply with our journal policies and formatting style in order to maximise the accessibility and therefore the impact of your work.

EDITORIAL REQUESTS

* Your manuscript should comply with our policies and format requirements, detailed in our style and formatting guide (<https://www.nature.com/documents/commsj-phys-style-formatting-guide-accept.pdf>).

* Please edit your manuscript according to the editorial requests in the attached table, and outline revisions made in the right hand column. If you have any questions or concerns about any of our requests, please do not hesitate to contact me. It is important that each request be addressed in order to avoid delays in accepting your manuscript. Please upload the completed table with your manuscript files as a Related Manuscript file.

* The editorial requests table also includes a full list of the files that must be provided upon resubmission. Please upload your files according to this table.

* An updated editorial policy checklist that verifies compliance with all required editorial policies must be completed and uploaded with the revised manuscript. All points on the policy checklist must be addressed; if needed, please revise your manuscript in response to these points. Please note that this form is a dynamic 'smart pdf' and must therefore be downloaded and completed in Adobe Reader. Clicking this link will download a zip file containing the pdf.

OPEN ACCESS

Communications Materials is a fully open access journal. Articles are made freely accessible on publication under a CC BY license (Creative Commons Attribution 4.0 International License). This license allows maximum dissemination and re-use of open access materials and is preferred by many research funding bodies.

For further information about article processing charges, open access funding, and advice and support from Nature Research, please visit <https://www.nature.com/commsmat/about/open-access>

RESUBMISSION

At acceptance, you will be provided with instructions for completing this CC BY license on behalf of all authors. This grants us the necessary permissions to publish your paper. Additionally, you will be asked to declare that all required third party permissions have been obtained, and to provide billing information in order to pay the article-processing charge (APC).

Please use the following link to submit your revised files:

[link redacted]

We hope to hear from you within two weeks; please let us know if the process may take longer.

Best regards,

Dr John Plummer

Chief Editor

Communications Materials